# Review of Pharmacotherapy for Tinnitus

**DOI:** 10.3390/healthcare9060779

**Published:** 2021-06-21

**Authors:** Sang Hoon Kim, Dokyoung Kim, Jae-Min Lee, Sun Kyu Lee, Hee Jin Kang, Seung Geun Yeo

**Affiliations:** 1Department of Otorhinolaryngology, Head and Neck Surgery, School of Medicine, Kyung Hee University, Seoul 02447, Korea; hoon0700@naver.com (S.H.K.); sunjaesa@hanmail.net (J.-M.L.); pseudoi@hanmail.net (S.K.L.); gmlwlskkk@naver.com (H.J.K.); 2Department of Anatomy and Neurobiology, College of Medicine, Kyung Hee University, Seoul 02447, Korea; dkim@khu.ac.kr

**Keywords:** tinnitus, pharmacotherapy

## Abstract

Various medications are currently used in the treatment of tinnitus, including anesthetics, antiarrhythmics, anticonvulsants, antidepressants, antihistamines, antipsychotics, anxiolytics, calcium channel blockers, cholinergic antagonists, NMDA antagonists, muscle relaxants, vasodilators, and vitamins. To date, however, no medications have been specifically approved to treat tinnitus by the US Food and Drug Administration (FDA). In addition, medicines used to treat other diseases, as well as foods and other ingested materials, can result in unwanted tinnitus. These include alcohol, antineoplastic chemotherapeutic agents and heavy metals, antimetabolites, antitumor agents, antibiotics, caffeine, cocaine, marijuana, nonnarcotic analgesics and antipyretics, ototoxic antibiotics and diuretics, oral contraceptives, quinine and chloroquine, and salicylates. This review, therefore, describes the medications currently used to treat tinnitus, including their mechanisms of action, therapeutic effects, dosages, and side-effects. In addition, this review describes the medications, foods, and other ingested agents that can induce unwanted tinnitus, as well as their mechanisms of action.

## 1. Introduction

The incidence of tinnitus, one of the most frequent symptoms in otology, has tended to increase over time. This increased incidence has been ascribed to developments in modern society, including drug abuse, noise pollution, aging of the population, deterioration due to stress and overwork, the increased incidence of chronic diseases, and easy access to medicines [1]. Because of the heterogeneity of tinnitus, a single hypothesis or theory is insufficient to explain the mechanism of its development. Degeneration of outer hair cells in the peripheral auditory system has been associated with tinnitus, whereas the auditory plasticity theory, including the upregulation of excitation of central auditory structures, can explain the role of the central auditory pathway in the generation of tinnitus. Other hypotheses associated with the pathophysiology of tinnitus include changes in biochemical systems, discordant dysfunction, spontaneous hyperactivity of the auditory tract, and imbalances between inhibitory and excitatory transmitter activities of the central auditory tracts. Hypotheses related to neural plasticity and cortical reorganization have also recently been introduced. In addition, the somatosensory and limbic autonomic nervous systems have been associated with the pathogenesis of tinnitus [2,3,4]. To date, a distinct cause or mechanism of development has not been identified, making tinnitus difficult to precisely diagnose and treatment.

Various methods are currently used to treat tinnitus, including medications, cognitive behavioral therapy, neurobiofeedback, neuromodulation, tinnitus retraining therapy, sound therapy, and hearing aids. Although medications are frequently used in primary care, no medication has been approved as a treatment for tinnitus by the US Food and Drug Administration [5]. Several of these medicines are thought to prevent the physiological and pathological changes associated with tinnitus, whereas others help patients more effectively endure tinnitus. In addition, medications that help patients overcome depression, anxiety, and insomnia have been used to treat tinnitus [6]. Understanding the mechanisms of action of these medicines, including their in vivo pharmacological effects and side-effects, is very important in treating and controlling tinnitus. Factors that should be considered in selecting medicines to treat tinnitus should include their dosage, period of use, side-effects, chemical dependence, possible additive effects, withdrawal symptoms, drug resistance, and effects on habituation. This review, therefore, describes the medicines often used to treat tinnitus, including their mechanisms of action and side-effects.

The PubMed, ScienceDirect, and the Cochrane Center Register of Controlled Trials databases were searched, using the search terms pharmacotherapy or drug therapy for tinnitus. The reference lists of eligible studies were also reviewed. Studies were included if they met all three of the following inclusion criteria: (a) prospective investigative study; (b) all patients diagnosed with idiopathic subjective tinnitus; (c) published in English. Studies were excluded if they comprised (a) unpublished data, (b) grey literature, (c) case reports, or (d) duplicates of published research. Institutional review board approval and informed consent were not required for this review article. (Figure 1).

## 2. Medications for Tinnitus

Although no medicines have been approved to date by the FDA, several classes of medication are often used to treat patients with tinnitus (Table 1). These medicines can be classified by their sites of action, including those that directly or indirectly control responses within the cochlea; those that affect responses of the nervous system; those simultaneously acting on both the cochlea and nervous system; and others.

### 2.1. Anesthetics and Antiarrhythmics

#### 2.1.1. Lidocaine

Lidocaine, which reduce abnormal spontaneous hyperactivity in the central nervous system, is often used to treat tinnitus. Lidocaine-associated inhibition of tinnitus is consistent with the concept of sensory epilepsy, with local anesthetics having inhibitory effects on the central nervous system and alleviating abnormal hypersensitivity. In addition, lidocaine was found to improve blood flow in the internal ear and simultaneously reduce cochlear microphonic and action potential by extending vessels [7].

Initially, insertion of procaine into the nasal concha was reported to reduce tinnitus [8]. Intravenous injection of lidocaine into 78 patients with tinnitus resulted in its complete disappearance in 27 patients (35%), its partial improvement in 22 patients (28%), and no effect in 21 patients (26%) [9,10]. In a placebo-controlled, double-blind study, tocainide HCl administered four times a day resulted in 80% to 98% tinnitus relief in five of the six patients who tolerated the drug [11]. Majumdar et al. observed significant improvements in tinnitus in 13 of 20 patients treated with lignocaine compared with only 3 of 19 patients treated with saline, with significant reductions in the amplitude of the action potential after lignocaine injection [12]. Intravenously injected lidocaine should be considered if conservative treatment for tinnitus is not effective. Lidocaine, however, should not be administered to patients aged <18 years, women of childbearing age, or patients with hyper-reactivity to the amide line of local anesthetics, as well as those with first degree ECG block, renal failure, or acute abnormal liver function [13,14].

The side-effects of lidocaine include sleepiness, anxiety, nausea, vomiting, speech disorders, numbness, and visual dimness, with the most serious side-effects of lidocaine being shock symptoms, including lower blood pressure, pulse abnormality, breathing inhibition, and cramps. Patients or their caregivers should be provided adequate information about the effects and side-effects of injectable lidocaine and should provide signed informed consent forms prior to treatment [15,16] (Table 2).

#### 2.1.2. Iontophoresis

In iontophoresis, the external auditory canal is filled with lidocaine by applying a solution of ionized materials and inserting them into tissues, followed by the repeat application of electric stimuli of 0.7–2 mA for 10 min per day for five days, depending on the patient response. A study in 50 patients showed that although none experienced complete disappearance of tinnitus, 30 (60%) showed improvements. Moreover, iontophoresis resulted in greater improvements in tinnitus than placebo [17].

#### 2.1.3. Tocainide Hydrochloride

Tocainide hydrochloride was initially developed to treat arrhythmia and to overcome the drawbacks of lidocaine. Lidocaine, which must be injected intravenously, has a short half-life of several minutes, whereas tocainide hydrochloride is administered orally, has a half-life of 11 h, and has fewer side-effects than lidocaine [18]. Tocainide hydrochloride, however, is effective only in some patients with subjective idiopathic tinnitus and is less effective than lidocaine. Its side-effects may include nausea, skin rash, digestive disorder, and myelosuppression [19,20,21,22] (Table 3).

### 2.2. Anticonvulsants

Anticonvulsants have been used to treat tinnitus, based on the hypothesis that tinnitus was associated with hyperactivity of the auditory system. Anticonvulsants may reinforce the inhibition of the central auditory system by enhancing the activity of gamma-aminobutyric acid (GABA), an inhibitory neurotransmitter. They may also alleviate the level of excitement by reducing the concentration of glutamate, an excitatory neurotransmitter in the auditory system. In addition, anticonvulsants may suppress the depolarization and central activation of cells by blocking voltage-dependent sodium channels [23].

#### 2.2.1. Carbamazepine

Carbamazepine is an oral anticonvulsion and antidepression agent that may control tinnitus. Because tinnitus is an indicator of pain induced by abnormal hypersensitivity, carbamazepine may be as effective as lidocaine, suggesting that carbamazepine would be effective for patients who have benefited from lidocaine treatment. The carbamazepine dosage is usually increased slowly, from 100 mg three times per day to 600–1000 mg per day. Its side-effects include dizziness, stomach disorders, skin rash, leucopenia, low blood pressure, abnormal liver function, and blood disorders.

A significant benefit from carbamazepine has not been confirmed in placebo-controlled studies of tinnitus patients [24,25,26]. Other studies, however, have found that this agent was effective in patients with intermittent tinnitus, for example when the pattern consists of a typewriter, popping corn, or ear clicking sound [27] (Table 4).

#### 2.2.2. Gabapentin

As an inhibitory neurotransmitter acting on voltage-gated calcium channels, the role of gabapentin in the treatment of tinnitus remains unclear. Gabapentin, however, has been reported to significantly improve annoyance and loudness of tinnitus related to acoustic trauma. It was also reported that the combination of gabapentin and lidocaine was superior to placebo. Other studies found no differences in either annoyance or loudness of tinnitus between gabapentin and placebo [28,29,30,31]. Although gabapentin may improve tinnitus caused by acoustic trauma, there is insufficient evidence supporting the effectiveness of gabapentin in the treatment of tinnitus (Table 5).

#### 2.2.3. Phenytoin Sodium

Although phenytoin sodium has been reported to be more effective than carbamazepine, other studies have suggested that phenytoin sodium is less effective. Patients initially responsive to carbamazepine but who have to be discontinued due to its side-effects may benefit from treatment with phenytoin sodium, whereas other patients may benefit from treatment with both. The dosage of phenytoin sodium is usually increased from 30 mg three times per day to 100 mg per day. At maximal dosage, however, the concentrations of phenytoin sodium should be measured in blood and leukocytes [32].

#### 2.2.4. Sodium Valproate

Sodium valproate is an anticonvulsant that increases the levels of GABA in brains. Sodium valproate acts as a chlorine channel blocker in patients unresponsive to or intolerant of lidocaine, carbamazepine, phenytoin, or tocainide, but it has a weaker effect than carbamazepine. Its dosage is usually increased from 200 mg three times per day to 400 mg four times per day [33].

#### 2.2.5. Primidone

Primidone is an anticonvulsant effective against tertiary neuralgia. Its dosage is gradually increased from 250 mg two times per day to 2 g per day. One trial reported that 27% of patients experienced 80–100% improvement in tinnitus and 59% experienced 20–80% improvement [15]. The side-effects of primidone include serious sedation, dizziness, ataxia, and anemia, with no significant differences in effects or side-effects between primidone and carbamazepine [34].

### 2.3. Antidepressants

Depression has been observed in 80% of patients with serious tinnitus, suggesting a correlation between tinnitus and depression. Thus, antidepressants can help relieve the stress, anxiety, and depression related to tinnitus, minimizing the psychological burden of the condition. Amitriptyline inhibits the re-absorption of serotonin and norepinephrine, nortriptyline inhibits the re-absorption of norepinephrine, while trimipramine inhibits the postsynaptic receptors of dopamine and serotonin [35].

A double-blind, placebo-controlled trial in patients with serious tinnitus accompanied by depression or depressive symptoms found that the tricyclic antidepressant (TCA) nortriptyline significantly reduced depression scores, tinnitus disability scores, and tinnitus loudness [36]. Nortriptyline, however, had no effect on tinnitus in non-depressed patients, and may even induce the development of tinnitus in these individuals. The side-effects of tricyclic and heterocyclic antidepressants include orthostatic hypotension, tachycardia, nausea, vomiting, sedation, and extrapyramidal symptoms. The side-effects of MAO inhibitors include hypertension, fever, arrhythmia, xerostomia, constipation, dizziness, and insomnia. Four double-blind, placebo-controlled studies reported that high dosages of TCAs and selective serotonin reuptake inhibitors (SSRIs) are effective for tinnitus accompanied by depression and anxiety or insomnia. The SSRIs paroxetine and sertraline were found to have an effect on tinnitus. Other SSRIs used to treat patients with tinnitus include fluoxetine, fluvoxamine, citalopram, and escitalopram [37,38,39]. Although sertraline was reported to be more effective than placebo in patients with serious incurable tinnitus, tinnitus may recur in these patients following the discontinuation of sertraline [40]. In contrast, trazodone was not effective in the treatment of tinnitus [41] (Table 6). Although a systematic review of the literature indicated that antidepressants may have effects in patients with tinnitus, the benefits of these agents remain unclear [42].

### 2.4. Antihistamines

Antihistamines act by competitively inhibiting peripheral and central histamine receptors. Antihistamines contribute to the treatment of factors affecting the onset of tinnitus. For example, their drying effect improves the functions of the auditory tubes, reduces the accumulation of effusions in the middle ear, and enhances ventilation in the tympanic cavities. Tinnitus results from continuous physical pressure reaching the cochlea. By reducing pressure in the tympanic cavity, antihistamines may affect the internal ears through oval and round windows, displacing the tectorial membranes associated with hair cells [43] and reducing the incidence of tinnitus. In addition, antihistamines may be effective for endolymphatic hydrops, as they have anticholinergic effects, strongly relaxing the capillaries and venules by affecting even the cochlear vessels. Betahistine hydrochloride, first used in Europe in 1970 to treat patients with Meniere’s syndrome, was found to increase cerebral blood flow and blood flow of the inner ears. Although several studies have indicated that betahistine may be effective in the management of tinnitus, there is insufficient evidence to suggest that betahistine has an effect on Meniere’s disease or subjective idiopathic tinnitus [44,45]. Other antihistamines, including terfenadine, chlorpheniramine, and meclizine, have been used to treat patients with tinnitus accompanied by allergies [46]. Primary side-effects include stimulus or inhibition of the central nervous system, anxiety, and sedation.

### 2.5. Anxiolytics

#### 2.5.1. Benzodiazepines

Benzodiazepines (BDZs) and other types of anxiolytic agents are among the medicines most often prescribed for tinnitus. BDZs, which bind to GABA receptors and reinforce GABA activity, inhibit the extension of tinnitus to acoustic centers, resolving the anxiety and depression caused by tinnitus and promoting sleep [47].

The side-effects of BDZs include dependency during their long-term use, sedation, ataxia, and depression. These agents, however, carry a risk of misuse or abuse, as well as having central nervous side-effects, such as drug dependence, personality changes, memory failure, and sedation, limiting their use in many patients. In general, medicines with shorter half-lives exhibit better effects, although they are characterized by higher rates of dependence [48].

Compared with a control placebo group, treatment with 30 mg per day of oxazepam significantly improved tinnitus in 52% of patients, whereas treatment with 0.5 mg of clonazepam three times per day significantly improved tinnitus in 69% of patients. Clonazepam and oxazepam may control tinnitus by inhibiting centripetal signals from the brain stem or the central nervous system above them [26,49,50]. BDZs, however, may have deleterious effects in the treatment of patients with tinnitus, in that diazepam may worsen depression and clonazepam may not inhibit tinnitus but rather induce tolerance to it [51].

A placebo-controlled study found that three BDZs, alprazolam, midazolam, and clonazepam, had significant therapeutic effects [52]. BDZs with longer half-lives, such as diazepam and oxazepam but not clonazepam, were reported to have no therapeutic effects, with delayed tinnitus reported after their discontinuation (Table 7). Compared with patients receiving placebo, those treated with clonazepam showed significant improvements in tinnitus measured on a visual analog scale. Another study showed that clonazepam significantly improved tinnitus loudness (74% of subjects), duration (63%), annoyance (79%), and tinnitus handicap inventory score (61%). Moreover, clonazepam is safe, with only a small likelihood of misuse, due to its relatively short half-life [53,54,55,56,57]. A systematic review of the literature found that the benefits of most BDZs were unclear in patients with subjective tinnitus [48].

#### 2.5.2. Barbiturates

Barbiturates induce different levels of sedation, hypnosis, and anesthesia, depending on their dosages and rates of administration. Reductions in cerebral blood flow and cerebrospinal fluid pressure result in reduced intracranial pressure, leading to reductions in cerebral oxygen consumption and metabolism, indicating that barbiturates may be useful in treating brain injuries caused by hypoxemia. Because barbiturates have anticoagulant activities, they are contraindicated for patients with acute porphyria. Although several studies have reported that barbiturates had therapeutic effects in 92% of patients with tinnitus, other studies found that barbiturates were ineffective [58,59] (Table 8).

### 2.6. Calcium Channel Blockers

Calcium concentrations inside and outside cochlear cells may contribute to the development of tinnitus. One study reported that treatment with 90 mg/day nimodipine for 3 weeks improved tinnitus in 20 (67%) of 30 patients, whereas another study reported that that only five (16%) of 31 patients significantly benefited from nimodipine, whereas two (6%) patients experienced worsening of tinnitus, indicating that nimodipine has contradictory outcomes in patients with tinnitus. Side-effects of nimodipine may include low blood pressure, edema, headache, tachycardia, skin flare, phlebitis, and hepatic failure [60].

### 2.7. Diuretics

Furosemide is a loop diuretic that inhibits the Na-K-2Cl cotransporter and endolymphatic potential. Although i.v. furosemide has no effect on central tinnitus, furosemide selectively suppresses tinnitus of peripheral etiology [61].

Intravenous injections of furosemide improved symptoms in 50% of patients with tinnitus and 40% of patients with Meniere’s disease. Caution should be exercised, however, when administering high doses of furosemide, because this agent may induce temporary hearing loss and tinnitus [62].

### 2.8. NMDA Antagonists: N-Methyl-D-Aspartate Receptor Antagonist

Cochlear tinnitus may arise from excitotoxicity via NMDA receptors. An imbalance between excitatory and inhibitory neurotransmitters was observed in some parts of auditory pathways in patients with tinnitus. High dosages of salicylate increase nerve conduction through NMDA receptors in the cochlear spiral ganglions, resulting in tinnitus. This effect disappears following administration of NMDA antagonists, suggesting that salicylate increases the glutamate-induced irritability of auditory nerves, an effect likely caused by NMDA receptors [1].

Caroverine has therapeutic effects on glutamate-mediated neurotoxicity of the cochlea by acting as an agonist for glutamate receptors in the cochlea. A single intravenous injection of caroverine was found to immediately alleviate tinnitus in 63% of patients, with 43% of those who showed improvement continuing to show improvement one week after treatment [63]. In contrast, a second study using the same design found that caroverine did not demonstrate any significant therapeutic effects [64]. Memantine is a voltage-dependent antagonist against NMDA receptors that decreases excitotoxicity by reducing calcium influx, which has been approved for treatment of patients with Alzheimer’s disease [65].

Acamprosate is an agent that controls GABAergic or glutamatergic transmission by blocking NMDA receptors and increasing GABAergic transmission. This medicine, which has been approved by the US FDA for treating alcoholism, was expected to be effective in treating tinnitus, as the reductions of GABAergic inhibition and glutamatergic excitation were associated with the onset of tinnitus. A randomized clinical trial showed a high index of success for tinnitus relief of 86.9%, with >50% relief observed in 47.8% of cases [66]. A double-blinded, placebo-controlled trial of patients with noise-induced tinnitus found that patients treated with 333 mg acamprosate three times per day for more than 90 days showed statistically significant improvements in tinnitus scores, compared to only 12.5% of patients receiving placebo [67]. Neramexane is a recently introduced drug that exhibits antagonistic properties in α9α10 cholinergic nicotinic receptors and N-methyl-D-aspartate receptors, suggesting potential efficacy in the treatment of tinnitus. Four weeks after the end of treatment with this drug, THI scores in the 50 mg/d group were significantly better than those of controls [68] (Table 9).

### 2.9. Muscle Relaxants

As a selective GABA analogue to GABA receptor, baclofen, a GABA agonist, is used to treat tertiary neuralgia and is effective in increasing muscle tension. L-baclofen had a beneficial effect on cochleovestibular compression syndrome in an animal model, as it effectively inhibited cochlear nucleus and inferior colliculus, as well as tinnitus. A double-blinded, placebo-controlled trial showed that tinnitus was improved in 1 of 32 patients (3.4%) administered placebo and 3 of 31 (9.7%) administered baclofen, but this difference was not statistically significant. Side-effects of baclofen included confusion, dizziness and gastrointestinal disorder [69]. The finding of a correlation between an increased intensity of tinnitus and increased muscle tension suggested that eperisone hydrochloride may be effective in patients with tinnitus [70].

### 2.10. Vasodilators

Because tinnitus may result from disturbances in cochlear and cerebral blood flow disturbances, vasodilators may be effective in its treatment. Treatment with vasodilators was found to improve symptoms in 47% of patients with sudden hearing loss. Moreover, vasodilators were shown to be useful in treating blood circulatory disturbances in the internal ears, with regulation of blood flow in the internal ears and cerebrovascular automatic nerve regulation being related to variations in blood pressure. In a double-blind trial of 50 patients with tinnitus due to various causes, no difference was observed between flunarizine and the placebo with respect to the suppressive effect on tinnitus [71,72,73].

### 2.11. Vitamins

Cochlear function depends on adequate vascular supply and the normal functioning of nerve tissue. Vitamin B12 deficiency is associated with axonal degeneration, demyelination, and subsequent apoptotic neuronal death, and may cause the demyelination of neurons in the cochlear nerve, resulting in hearing loss. In addition, low levels of vitamin B12 and folate are associated with the destruction of the microvasculature of the stria vascularis, which might result in decreased endocochlear potential and in hearing loss and tinnitus.

A pilot study reported a high prevalence of vitamin B12 deficiency in a North Indian population and improvements in tinnitus severity scores and VAS in cobalamin-deficient patients receiving intramuscular vitamin B12 weekly for 6 weeks [74]. In contrast, vitamin B12 replacement treatment was not effective in 100 patients with non-pulsatile tinnitus and vitamin B12 deficiency [75].

Nicotinic acid is a water-soluble vitamin found in meats, milk, eggs, and other foods. It is necessary to maintain skin, nerves, and digestive function, and its deficiency may result in pellagra. Nicotinic acid is used to control tinnitus, as it enhances blood flow in the labyrinth. For example, 50% of patients with tinnitus were reported to benefit from nicotinic acid administration [76]. In a double-blind, controlled trial performed in 48 patients, the value of the related drug nicotinamide was assessed and compared with the effect of a placebo. The results obtained with nicotinamide were not better than those observed for placebo [77]. Its side-effects included gastric ulcers, hypotension, headache, and hepatic failure.

### 2.12. Others

#### 2.12.1. ATP (Adenosine Triphosphate)

Hydrolysis of ATP in vivo has been reported to partially alleviate tinnitus through the generation of energy. In other parts of the body, ATP hydrolysis was found to increase cerebrovascular resistance and improve peripheral symptoms [78].

#### 2.12.2. Atorvastatin

Atorvastatin decreases blood cholesterol and prevents vascular events by inhibiting HMG-CoA reductase and reducing cholesterol synthesis. A randomized clinical trial of elderly patients with tinnitus and high cholesterol found that treatment with atorvastatin for 13 months did not delay the development of age-related hearing loss or significantly reduce tinnitus [79]. Although several studies have reported that statin treatment significantly improved tinnitus, other studies have found that statins did not significantly improve this condition; thus, the benefits of statins in patients with tinnitus remain unclear [80].

#### 2.12.3. Dopaminergic and Antidopaminergic Agents

Dopamine acts as an inhibitory transmitter in the cochlea, whereas antidopaminergic agents enhance the thalamic filtering of sensory signals. Because the dopaminergic pathways in the limbic systems and prefrontal areas are associated with the emotional aspect of tinnitus, dopaminergic agents and antagonists have been used to treat tinnitus. Simultaneous treatment with sulpiride plus melatonin or hydroxyzine, however, resulted in significantly greater reductions in both the tinnitus and tinnitus perception VAS scores than administration of placebo. In contrast, piribedil, a D2/D3 agent currently used to treat patients with Parkinson’s disease, had no effect on tinnitus [81].

#### 2.12.4. Ginkgo Biloba Extract

Ginkgo biloba extract has a mechanism of action associated with neurotransmitters involved in blood circulation, metabolism, blood viscosity, and aging. Ginkgo biloba extract increases the permeability of capillaries and venous blood pressure and enhances ATP and blood glucose concentrations following metabolic damage to cerebral metabolism. In addition, ginkgo biloba extract reduces platelet agglutination and affects the acetocholinergic system by acting as a neurotransmitter. Treatment usually consists of administration of about 100 mg/kg per day for about 3 weeks. Some studies have suggested that ginkgo biloba extract affects nucleus-derived and peripheral tinnitus, whereas others do not [82,83,84] (Table 10). If ginkgo biloba extract increases the risk of bruising and bleeding in a patient scheduled for surgery, its use should be discontinued 2 weeks before the operation. Other side-effects include constipation, dizziness, forceful heartbeat, headache, and stomach upset. Although a number of reports have indicated that ginkgo biloba may be effective in the management of tinnitus, the benefits of this agent remain unclear [85].

#### 2.12.5. Melatonin

Melatonin controls circadian rhythms through a combination of circadian hormones that occur naturally with melatonin receptors. As a strong oxidizing agent, melatonin has been shown to prevent noise- and medicine-induced tinnitus by protecting mitochondrial and nuclear DNA. Insomnia is one of the main symptoms in patients with tinnitus, as well as being a cause of tinnitus, with melatonin being effective in treating insomnia [86]. Prospective clinical trials of the treatment of tinnitus patients with 3.0 mg/day melatonin found that melatonin was associated with a statistically significant decrease in tinnitus intensity and improved sleep quality in patients with chronic tinnitus [87,88,89] (Table 11).

#### 2.12.6. Pentoxifylline

The xanthine derivative pentoxifylline enhances blood flow by reducing its viscosity.

A study of 30 patients treated with 3 × 400 mg pentoxifylline orally observed significantly increased cerebral blood flow and significantly improved tinnitus during the pentoxifylline observation period [90]. In another study, however, no significant difference was observed between the placebo and oxpentifylline treatment groups for any of the parameters measured [91] (Table 12).

#### 2.12.7. Prostaglandin E1

Aspirin and other non-steroidal anti-inflammatory drugs (NSAIDs) that inhibit cyclooxygenase have been found to reduce prostaglandin levels, resulting in tinnitus and temporary hearing loss; thus, by inhibiting platelet agglutination and extending vessels, thereby improving internal ear circulation, prostaglandin E1 may be effective in treating tinnitus. Treatment with 200 mg/day prostaglandin E1 with increases in dosage every 5 days was found to improve tinnitus in 33% of patients [92].

#### 2.12.8. Sodium Fluoride

Sodium fluoride (NaF) has been used to treat sclerosis of the spinal cord and cochlea and has begun to be used to treat otosclerosis, based on the hypothesis that NaF can prevent the development of sensorineural hearing loss by enhancing the maturity of otospongiosis lesions. NaF was reported to control the sites of otospongiosis and alleviate tinnitus. Moreover, administration of 40 mg/day NaF plus vitamin D was found to have beneficial effects on both otospongiosis and tinnitus [93].

#### 2.12.9. Steroids: Intratympanic Steroid Injection

Intratympanically injected steroids are used to treat tinnitus, as they alleviate irritability of the tympanic plexus and improve hypersthenia of the internal ears. Intratympanic injection of dexamethasone into 50 patients with subjective tinnitus three times per day for 2 weeks resulted in complete and partial improvements of symptoms in 17 (34%) and 20 (40%) patients, respectively, with the other 13 (26%) showing no change. In another trial, 139 patients with acute tinnitus occurring during the previous 3 months were treated with intratympanic steroids, with 37.7% reporting improvements in tinnitus 3 months later [94,95,96,97,98] (Table 13).

#### 2.12.10. Trimetazidine HCl

Trimetazidine HCl inhibits the generation of free radicals noxious to cells by directly preventing acidification in ischemic cells and promoting the generation of ATP, a source of energy. This agent is also used to treat heart attacks, ischemic chorioretinal disorders, and vascular dizziness, as well as tinnitus. Treatment of 21 patients with tinnitus for 60 days reduced the strength and frequency of tinnitus by 52.4% and 47.6%, respectively (Table 14). Side-effects of trimetazidine HCl include headache, rash, nausea, and lack of appetite [99,100].

#### 2.12.11. Zinc

Many cellular metabolic processes require zinc, with this metal ion playing important roles in growth and development, immune responses, neurological functions, and reproduction. Zinc has been found to catalyze chemical responses necessary for maintaining life, to be involved in the structure of proteins and cell membranes, and to regulate gene expression by acting as a transcription factor.

Zinc concentrations are higher in the cochlea and vestibular sites than in other parts of the body, suggesting that zinc may be effective in treating tinnitus. Zinc deficiency is reported to induce senile hearing loss, whereas administration of zinc improved hearing loss and tinnitus in one-third of elderly patients with low zinc concentrations. Zinc deficiency is associated with tinnitus and sensorineural hearing loss, as well as with nail fragility, depilation, taste disorders, olfactory disturbances, nyctalopia, and prostate diseases. The blood concentration of zinc is inversely proportional to the concentration of copper. Good outcomes have been reported in patients with tinnitus treated with 150 mg/day zinc and 600 mg/day zinc sulfate, amounts ten times higher than required by normal adults. Serum zinc concentrations differed significantly between control subjects and patients with tinnitus, with the latter showing improvement following oral administration of 34‖68 mg/day zinc for 2 weeks. Zinc, however, was ineffective in tinnitus patients with normal zinc concentrations [101,102].

## 3. Medicines and Materials That Exacerbate Tinnitus

Treatment with certain medicines or ingestion of certain foods may result in unexpected tinnitus. In general, ototoxic medicines associated with tinnitus include salicylate, analgesics, NSAIDs, aminoglycoside antibiotics, quinine-containing antimalarial drugs, and loop diuretics. Other drugs, wines, and food have also been found to cause or exacerbate tinnitus (Table 15).

### 3.1. Alcohol

Alcohol inhibits the central nervous system. Although non-alcoholic patients with tinnitus may benefit from alcohol, most patients report an increase in the strength of tinnitus, suggesting that patients with tinnitus should avoid alcohol [103].

### 3.2. Anticancer Drugs and Heavy Metals

Subjective tinnitus is a common side-effect of heavy-metal-containing anticancer medicines, including alkylating agents, such as nitrogen mustard, cyclophosphamide, melphalan, and chlorambucil; antimetabolites, such as methotrexate, pyrimidine, and 5-fluorouracil; antitumor antibiotics, such as dactinomycin, doxorubicin, bleomycin, and mitomycin; and cisplatin.

Cisplatin-induced ototoxicity may be exacerbated by fosfomycin, a phosphonic acid antibiotic that can counteract the effects of cochleotoxic and nephrotoxic medications. About 50% of patients administered 2 mg/m^2^ of cisplatin experienced a 15 dB hearing loss on pure tone audiometry, usually at frequencies of 6–8 kHz, with tinnitus occurring before audiographic abnormalities were observed. Combinations of cisplatin with anticancer drugs, such as 5-fluorouracil, bleomycin, methotrexate, and leucovorin, result in the development of tinnitus. In addition, long-term treatment with cisplatin is ototoxic, with recovery from tinnitus observed after the medication is halted [104]. Tinnitus was found to be more frequent in patients treated with cisplatin than with other agents. Furthermore, patients with tinnitus were more likely to have a lower quality of life [105].

### 3.3. Caffeine

Caffeine has a substantial effect on vestibulocochlear functions and is associated with the development of tinnitus and the exacerbation of anxiety. Caffeine correlates not only with the onset of tinnitus but also with its strength. Tinnitus symptoms can be improved within a few days or weeks after caffeine is stopped [106]. Although caffeine may improve tinnitus distress, a recent study found no association between high caffeine consumption and tinnitus distress [107]. Moreover, a systematic review reported that the effect of caffeine on the development or reduction of tinnitus was unclear [108].

### 3.4. Cocaine

Little evidence is currently available on a correlation between cocaine and the onset of tinnitus or its clinical natural history [109].

### 3.5. Nonnarcotic Analgesics and Antipyretics

Ibuprofen and naproxen are medications that can induce reversible hearing loss and tinnitus. A comparison of dosage per patient weight of ibuprofen and salicylate showed that the ototoxicity of the former was weaker than that of the latter. Cessation of treatment results in a recovery of hearing and reversal of tinnitus within a few weeks. Indomethacin may also induce reversible hearing loss and tinnitus [110].

### 3.6. Oral Contraceptives

Oral contraceptives may also be associated with the development of tinnitus and hearing loss, ototoxicities associated with the blood vascular system [111].

### 3.7. Ototoxic Antibiotics

Streptomycin; aminoglycosides such as amikacin, gentamicin, kanamycin, neomycin, tobramycin, netilmicin, and erythromycin; chloramphenicol; and colistimethate are antibiotics that are known to be ototoxic, and which may cause hearing loss, dizziness, tinnitus, and ear fullness, either alone or in combination with other agents. Topical antibiotics may also induce tinnitus. Aminoglycoside antibiotics may induce permanent hearing loss and permanent tinnitus, either alone or in correlation with loop diuretics such as ethacrynic acid. Correlations between aminoglycoside and non-aminoglycoside antibiotics, including viomycin, kapramycin, polymyxin B, and cis-platinum, may also cause cytoclasis in the cochlea and auditory disorders [112].

Vancomycin may cause permanent ototoxicity and high doses of erythromycin may result in temporary and reversible hearing loss and tinnitus. Doxycycline and minocycline frequently cause tinnitus [113].

### 3.8. Ototoxic Diuretics

Ethacrynic acid and furosemide are toxic to the cochlea and may induce hearing loss and tinnitus [114].

### 3.9. Propranolol

The beta-adrenergic blocking agent propranolol acts on conducting tissues, making them susceptible to tinnitus and increasing its intensity [115].

### 3.10. Quinine and Chloroquine

Administration of large dosages of the ototoxic medications quinine and chloroquine to patients susceptible to them may result in hearing loss and tinnitus. These effects on hearing may be temporary or permanent, and tinnitus may precede hearing loss. These agents may induce hearing loss and tinnitus by mechanisms involving vasoconstriction and cochlear blood flow. Patients can recover from hearing loss if treatment with these agents is stopped immediately after bradyacusia and tinnitus occur [116].

### 3.11. Salicylates

Salicylate (aspirin), a dosage-dependent ototoxic medicine, is associated with tinnitus and auditory disorders, with most patients recovering from hearing loss and tinnitus when treatment is discontinued. Doses of about 4.8 g/day cause hearing loss of 10–15 dB, with continuous treatment resulting in hearing loss of 40–50 dB. Although audiograms appear to be horizontal, higher vocal ranges are more affected than lower vocal ranges. Hearing loss is associated with reductions in serum salicylate concentrations, with recovery observed within 24–72 h [117].

The efficacy of salicylate therapy is monitored by measuring the occurrence of tinnitus. The most effective dosage of salicylate can be determined by verifying when tinnitus disappears while increasing or reducing its dosage. Salicylate-induced tinnitus occurs at frequencies of 7–9 kHz. In addition, the timing of bradyacusia, tinnitus, vertigo, and ear fullness after taking salicylate is unclear, although tinnitus has been shown to precede hearing loss [118].

## 4. Conclusions

Although various drugs are currently used to treat tinnitus, none has yet been approved for its treatment. Moreover, because of differences in the characteristics and symptoms of patients with tinnitus, it is difficult to determine which treatments are most successful.

All parts of the auditory nervous system close to the cochlea may be involved in the onset and exacerbation of tinnitus. Almost all central nervous system pathways are related to auditory pathways and the occurrence of tinnitus. These include the limbic system, which is responsible for memory and emotional control; the hypothalamus–pituitary–adrenal axis, which controls the release of stress hormones; the somatosensory cortex, which plays a role in neuroplasticity; and the auditory nervous system. Medicines for treating tinnitus must simultaneously control many other nervous system pathways; thus, tinnitus may be treatable not by a single medication but rather by a combination of several.

Although treatment may frequently be empirical, double-blinded trials are needed to determine the safety and efficacy of any medication. Agents shown to be effective in treating tinnitus include anticonvulsants, antianxiety medicines, antidepressants, antihistamines, antiarrhythmic agents, local anesthetics, vasodilators, tranquilizers, vitamin pills, and ginkgo biloba extracts. Because no single agent is completely effective, efforts are required to develop medications or combinations of medications that can cure tinnitus without side-effects.

The selection of a treatment for tinnitus should include consideration of the dosage and duration of treatment, the temporary and permanent side-effects of the drug, the possibility of drug dependence or addiction, withdrawal symptoms or tolerance, the effect on the normal habituation process, and whether the drug interferes with habituation training. In general, medications are administered to block the pathophysiologic mechanism that causes tinnitus and to help overcome depression, anxiety, and insomnia related to tinnitus. Consideration of the characteristics of tinnitus and identification of the accompanying diseases in individual patients can help select an appropriate medication for the treatment of tinnitus and its resulting anxiety and depression. Further determination of the pathophysiology of tinnitus and additional clinical studies may result in the development of safe and effective agents specifically designed to treat this condition.

## Figures and Tables

**Figure 1 healthcare-09-00779-f001:**
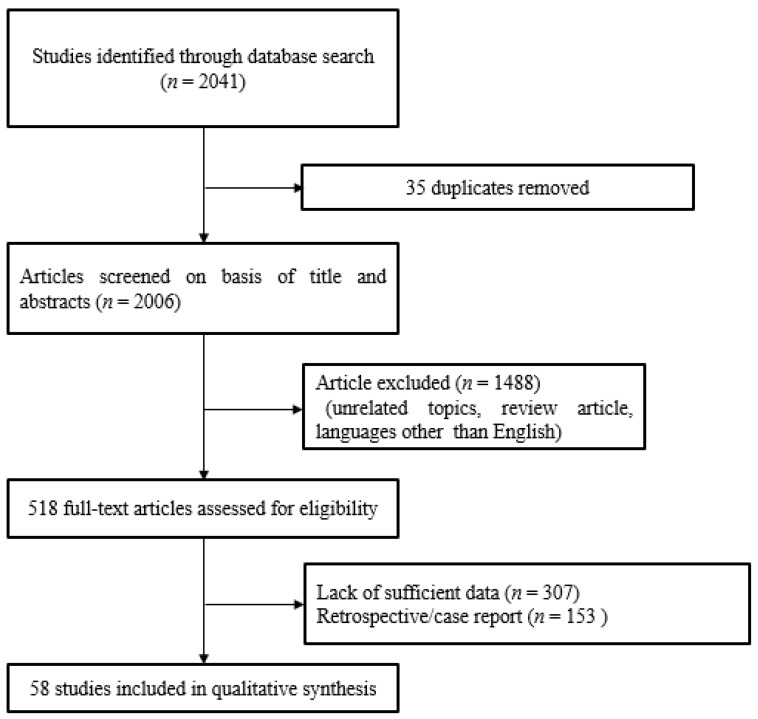
Flowchart of study selection according to PRISMA guidelines.

**Table 1 healthcare-09-00779-t001:** Off-label drugs used in the treatment of tinnitus.

Anesthetics	Antihistamines
Flecainide acetate	Chlorpheniramine
Lidocaine/lignocaine	Dexchlorpheniramine
Mexiletine—oral lidocaine analog	Meclizine
Procaine	Terfenadine
Tocaine—oral lidocaine analog	**Calcium Channel Blockers**
**Antiarrhythmics**	Flunarizine
Flecainide	Nifedipine
Lidocaine	Nimodipine
Mexiletine	**Diuretics**
Tocainide	Chlorothiazide
**Antianxiety**	Furosemide
Benzodiazepines	**Glutamate receptor antagonists**
Alprazolam	Acamprosate
Amylobarbitone	Caroverine
Clonazepam	Memantine
Diazepam	**Muscle relaxants**
Flurazepam	Baclofen
Oxazepam	Cyclobenzaprine
Protriptyline	Eperisone hydrochloride
**Anticonvulsants**	**Others**
Amino-oxyacetic acid	Aniracetam
Carbamazepine	Arlidin
Gabapentin	Atorvastatin
Lamotrigine	Clonidine
Phenytoin	Cyclandelate
Pregabalin	Glutamic acid diethylester
Primidone	Herbal products
Valproic acid	Melatonin
**Antidepressants**	Minerals
Amitriptyline	Naltrexone
Bupropion	Nicotinamide
Fluoxetine	Oxpentifylline/pentoxifylline
Nortriptyline	Cinnarizine
Paroxetine	Sulpiride
Sertraline	Vardenafil
Trimipramine	Vitamin

**Table 2 healthcare-09-00779-t002:** Summary of studies that used lidocaine to treat tinnitus.

Authors	Subjects	Success Rate	
		Relief	Lidocaine
Barany [8]	80	Relief of tinnitus temporarily (for 20 min up to 1 h)	

Emmett [11]	402 with unilateral tinnitus	50–100%	45%
	190 with bilateral tinnitus	50–100%	73%
Israel [15]	26	Better	22/26
		Worse	4/26
		No change	0/26
Duckert and Rees [7]	25	Improved	10/25
		Worsed	8/25
Majumdar [12]	20	Improved	13/20
		No improvement	7/20
Martin and Colman [10]	32	0%	7/32
		25%	2/32
		50%	4/32
		75%	5/32
		100%	14/32
Melding [9]	78	Total abolition	27/78
		Reduction	31/78
		No change	20/78
Perucca and Jackson [16]	5 with continuous tinnitus	None	0/5
		Improved	5/5

**Table 3 healthcare-09-00779-t003:** Summary of studies that used tocainide HCl to treat tinnitus.

Authors	Subjects	Success Rate	
		Relief	Tocainide HCl
Blayney [22]	32	Complete	1/32
		Partial	2/32
		No change	29/32
Cathcart [19]	26 with severe, constant,	Relief	6/26
	unilateral tinnitus	No change	17/26
		Increased (>50%)	3/26
Emma and Shea [18]	31	No difference between placebo and tocainide groups	
Hazell and Wood [20]	100 with severe subjective	Relief	4/10
	Tinnitus	No change	6/10
Hulshof and Vermeij [21]	19	Relief	8/19
		No change	11/19

**Table 4 healthcare-09-00779-t004:** Summary of studies that used carbamazepine to treat tinnitus.

Authors	Subjects	Success Rate	
		Relief	Carbamazepine
Donaldson [24]	62	Excellent	4/62
		Good	9/62
		Partial	15/62
		None	34/62
Hulshof and Vermeij [25]	24	Good	2/24
		No change	22/24
Lechtenberg and	19 treated	>80%	2/19
Shulman [26]	47 controls	50–80%	1/19
		<50%	2/19
		Worse	0/19
		No change	14/19
JS Han [27]	28complete	Complete	7/14
		Partial	3/14

**Table 5 healthcare-09-00779-t005:** Summary of studies that used gabapentin to treat tinnitus.

Authors	Subjects	Success Rate	
		Relief	Gabapentin
Bakhshaee [30]	16	Complete	2/16
		Partial	4/16
		None	10/16
Piccirillo [29]	59	Better	11/59
		Normal	40/59
		Worse	8/59
Witsell [28]	48	Better	18/48
		Same or Worse	30/48

**Table 6 healthcare-09-00779-t006:** Summary of studies that used antidepressants to treat tinnitus.

Drug	Authors	Subjects	Success Rate	
			Relief	Antidepressant
Amitriptyline	Bayar [38]	20	Success of treatment	19/20
	Podoshin [37]	76	Improvement	21/76
Nortriptyline	Sullivan [36]	120	Nortriptyline decreases depression, tinnitus related	
			disability and tinnitus loudness	
Paroxetine	Robinson [39]	50	Paroxetine was not statistically superior to placebo	
			on tinnitus measures	
Sertraline	Zoeger [40]	76	Sertraline was more effective than placebo in the	
			treatment of severe refractory tinnitus	
Trazodone	Dib [41]	43	Trazodone was not efficient in controlling tinnitus	
Trimipramine	Mihail [35]	26	Tricyclics were not beneficial	
			in the treatment of subjective tinnitus	

**Table 7 healthcare-09-00779-t007:** Summary of studies that used benzodiazepine to treat tinnitus.

Drug	Authors	Subjects	Success Rate	
			Relief	Benzodiazepine
Alprazolam	Johnson [54]	17	Reduction	13/17
	Jalali [56]	14	Alprazolam did not significantly improve the THI score or	
			sensation level of loudness	
Clonazepam	Bahmad [55]	10	Clonazepam reduced tinnitus annoyance and intensity	
	Han [57]	38	Improvement ≥ 2/3	21/38
			1/3 ≤ Improvement < 2/3	4/38
			Improvement < 1/3	3/38
			No change	10/38
	Lechtenberg [49]	26	>80%	4/26
			50–80%	8/26
			<50%	6/26
			Worse	0/26
			No change	8/26
Diazepam	Kay [53]	5	Not effective	
	Lechtenberg [26]	15	>80%	0/15
			50–80%	0/15
			<50%	1/15
			Worse	0/15
			No change	14/15
Flurazepam	Lechtenberg [26]	14	>80%	0/14
			50–80%	0/14
			<50%	2/14
			Worse	0/14
			No change	12/14
Oxazepam	Lechtenberg [26]	23	>80%	7/23
			50–80%	1/23
			<50%	4/23
			Worse	0/23
			No change	11/23

**Table 8 healthcare-09-00779-t008:** Summary of studies that used barbiturates to treat tinnitus.

Authors	Subjects	Success Rate		
		Relief	Before	After
Donaldson [58]	20	Only noticeable	3/20	9/20
		Always present	5/20	6/20
		Constant and disturbing	11/20	1/20
		Preventing sleep	1/20	0/20
		None	0/20	4/20
Mark [59]	9	No effect	9/9	
		Relief	0/9	

**Table 9 healthcare-09-00779-t009:** Summary of studies that used NMDA antagonists to treat tinnitus.

Drug	Authors	Subjects	Success Rate	
Acamprosate	Azevedo [66]	25	The beneficial effect increased from 1 month to 3 months	
Caroverine	Denk [63]	30	No improvement	11/30
			1 point better	14/30
			2 points better	4/30
			3 points better	1/30
	Domeisen [64]	24	Better	0/24
			Unchanged	20/24
			Worse	4/24
Memantine	Figueiredo [65]	43	Worsening (THI ≥ 10)	9/43
			Stable (−10 to 10)	18/43
			Improvement (THI ≤ −10)	16/43
Neramexane	Suckfüll [68]	317	Significantly better scores in the 50 mg/d group	

**Table 10 healthcare-09-00779-t010:** Summary of studies that used ginkgo biloba to treat tinnitus.

Authors	Subjects	Success Rate
Claussen [83]	16	Positive control in central origin tinnitus
Drew [82]	448	No more effective than placebo
Morgenstern [84]	59	Effective and safe in alleviating tinnitus

**Table 11 healthcare-09-00779-t011:** Summary of studies that used melatonin to treat tinnitus.

Authors	Subjects	Success Rate
Hurtuk [88]	61	Decrease in tinnitus and improved sleep quality
Megwalu [89]	24	Improved THI score and PSQI
Rosenberg [87]	30	Improved THI score and difficulty sleeping (14/30)

**Table 12 healthcare-09-00779-t012:** Summary of studies that used pentoxifylline to treat tinnitus.

Authors	Subjects	Success Rate	
Hartmann [90]	30	Tinnitus improved significantly	
Salama [91]	30	Improved	4/13
		No change	7/13
		worse	2/13

**Table 13 healthcare-09-00779-t013:** Summary of studies that used intratympanic steroid injection to treat tinnitus.

Authors	Subjects	Success Rate	
Cesarani [94]	50	Disappeared	17/50
		Significant decrease	20/50
		No improvement	13/50
Choung [98]	15	Not effective for refractory tinnitus	
Herraiz [95]	34	Tinnitus control in more than 70% of patients	
Shim [97]	35	ITD injection with alprazolam was a better; 75% improvement	
Topak [96]	70	No benefit with placebo	

**Table 14 healthcare-09-00779-t014:** Summary of studies that used trimetazidine to treat tinnitus.

Authors	Subjects	Success Rate	
Kumral [100]	42	No significant difference in THI, VAS, or subjective loudness score	
Rha [99]	21	Improved	11/42
		No improvement	29/24
		Worse	2/42

**Table 15 healthcare-09-00779-t015:** Drugs with tinnitus side-effects.

Acetazolamide	Diltiazem	Misoprostol
Acetylsalicylic acid	Erythromycin	Naproxen
Alprazolam	Etodolac	Netromycin
Aminoglycosides	Fluxetine	Nifedipine
Amitriptyline	Ibuprofen	Prazosin
amphotericin	Itraconazole	Piroxicam
Aztreonam	Ketorolac	Propofol
Barbiturates	Laratadine	Salicylates
Benzodiazepine	Lidocaine	Ticlopidine
Carbamazepine	Lisinopril	Triazolam
Diaminodiphenyl	Loop diuretics	Tobramycin
Diclofenac	Methotrexate	Vancomycine

## Data Availability

Not applicable.

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
