# Peer review of "Review of Pharmacotherapy for Tinnitus"

_healthcare, 2021, doi:10.3390/healthcare9060779_

Round 1

Reviewer 1 Report

The authors wrote a literary review about the medicines often
used to treat tinnitus, including their mechanisms of action and side effects.

The topic is always hot, because until now there are not guideline about the pharmacological treatment of tinnitus. 

The review is easy to read, interesting and well written. 

The article does not follow the standard guideline for a review. Please use these comments to improve the scientific quality of the article.

The introduction is too short. Please including the physiopathology and the possible etiology of tinnitus. For example the ear pathology, nervous pathology or other causes like temporomandibular dysfunction, please use these references:

  1. Flores-Orozco EI, Tiznado-Orozco GE, Díaz-Peña R, Orozco EIF, Galletti C, Gazia F, Galletti F. Effect of a Mandibular Advancement Device on the Upper Airway in a Patient With Obstructive Sleep Apnea. J Craniofac Surg. 2020 Jan/Feb;31(1):e32-e35. 
  2. Coelho CB, Santos R, Campara KF, Tyler R. Classification of Tinnitus: Multiple Causes with the Same Name. Otolaryngol Clin North Am. 2020 Aug;53(4):515-529. 
  3. Eggermont JJ, Roberts LE. The neuroscience of tinnitus. Trends Neurosci. 2004 Nov;27(11):676-82.

The review does not follow the PRISMA criteria, that are suggested in the guideline of the manuscript. So please add this flow-chart.

The review has needed of Discussion section, where you can summarize the most important concept and analyzed the most successful treatment.

Thanks for collaboration, hope to read your article with these changing.

Author Response

We thank this reviewer for the thoughtful and detailed review of our manuscript.

Reviewer 2 Report

Tab. 1: Stugeron. It's more appropriate to indicate the active principle (cinnarizine)

Why do you sometimes use the brand name (Stugeron, Dilantin, Mysoline, Misoprostol, Vastinan, tab. 15) of medications?

pag. 9: I suggest to consider the conclusions of 2 Cochrane databases. 1)There is insufficient evidence to say whether betahistine has any effect on Menière's disease. (James AL, Burton MJ. Betahistine for Menière's disease or syndrome. Cochrane Database Syst Rev. 2001;2001(1):CD001873. doi: 10.1002/14651858.CD001873.) 

2)There is an absence of evidence to suggest that betahistine has an effect on subjective idiopathic tinnitus when compared to placebo (Wegner I, Hall DA, Smit AL, McFerran D, Stegeman I. Betahistine for tinnitus. Cochrane Database Syst Rev. 2018 Dec 28;12(12): CD013093).

pag.11: updated reference: Prayuenyong P, Kasbekar AV, Baguley DM. The efficacy of statins as otoprotective agents: A systematic review. Clin Otolaryngol. 2020 Jan;45(1):21-31.

pag. 15: I suggest to cite this review on patients treated with cisplatin and hearing loss and tinnitus: Pearson SE, Taylor J, Patel P, Baguley DM. Cancer survivors treated with platinum-based chemotherapy affected by ototoxicity and the impact on quality of life: a narrative synthesis systematic review. Int J Audiol. 2019 Nov;58(11):685-695.

I suggest to consider a recent case-control study and a review on caffeine and tinnitus: Ghahraman MA, Farahani S, Tavanai E. A comprehensive review of the effects of caffeine on the auditory and vestibular systems. Nutr Neurosci. 2021 Apr 22:1-14. Figueiredo RR, Azevedo AA, Penido NO. Tinnitus features according to caffeine consumption. Prog Brain Res. 2021;262:335-344.

The authors didn't report if this review is conform to the relevant guidelines (PRISMA, Cochrane guidelines). If not it could be reported as a limit of the study.

The methods of the study are not clearly defined (search strategy? study selection criteria? data extraction? risk of bias assessment?) to analyze and report aggregated evidence on a pharmacotherapy for tinnitus. This is another limit of the study.

Author Response

(The authors gave the same response as above.)

Round 2

Reviewer 1 Report

With these corrections, the article is ready for pubblication.

Reviewer 2 Report

The corrections are appropriate.